# Destabilization and Ion Conductivity of Yttria-Stabilized Zirconia for Solid Oxide Electrolyte by Thermal Aging

**DOI:** 10.3390/ma15196947

**Published:** 2022-10-07

**Authors:** Hwanseok Lee, Kanghee Jo, Min-sung Park, Taewoo Kim, Heesoo Lee

**Affiliations:** 1School of Materials Science and Engineering, Pusan National University, Busan 46241, Korea; 2System & Energy Division Material Technology Center, Korea Testing Laboratory, Seoul 08389, Korea

**Keywords:** thermal aging, phase transformation, short-range ordering, oxygen vacancy, local atomic structure, ion conductivity

## Abstract

The degradation behavior of yttria-stabilized zirconia by thermal aging was investigated in terms of phase transformation, local atomic structure, and electrical conductivity. The average grain size of 8YSZ was increased from 20.83 μm to 25.81 μm with increasing aging temperature. All 8YSZ samples degraded at different temperatures had a predominantly cubic structure. The (400) peak of 8YSZ deteriorated at 1300 and 1400 °C shifted to a high angle, and the peak of tetragonal was not indexed. For 8YSZ degraded at 1500 °C, the (400) peak shifted to a lower angle, and the peak of tetragonal was identified. Analysis of the local microstructure of aged 8YSZ using extended X-ray absorption fine structure showed that the intensity of the Zr-O peak gradually increased and that the intensity of the peak of cationic Zr decreased as the aging temperature increased. The changes in the peaks indicate that the oxygen vacancies were reduced and Y^3+^ ions escaped from the lattice, leading to the destabilization of 8YSZ. The activation energies of 8YSZ at 1300 °C and 1400 °C were derived to be 0.86 and 0.87 eV, respectively, and the activation energy of 8YSZ at 1500 °C increased significantly to 0.92 eV. With the thermal deterioration of 8YSZ, the cation (Y^3+^) escaped from the lattice and the number of oxygen vacancies decreased, resulting in the formation of a tetragonal structure and high activation energy at 1500 °C.

## 1. Introduction

Oxygen sensors that operate under harsh conditions have been extensively developed in recent years, and zirconia electrolytes are in the spotlight because of their high ionic conductivity and excellent reliability [1,2]. Zirconia electrolytes are exposed to hazardous chemicals and extreme temperatures depending on the operating environment and operate in such environments for several years [3]. Long-term exposure to high temperatures causes degradation of the solid electrolyte, and the degradation changes the conductivity, affecting the accuracy and stability of the sensor [4]. In addition, the solid oxide membrane (SOM) process, which is an environmentally friendly metal electrolysis process using zirconia-based electrolyte as an oxygen-conducting membrane, has been studied recently [5,6]. Thus, ensuring the stability of zirconia-based electrolytes at high temperatures is becoming more important.

The electrical properties and stability of zirconia-based electrolytes depend on the dopant type and concentration [7]. Zirconia is a polymorphic oxide that is monoclinic at room temperature and transforms to tetragonal and cubic phases at approximately 1443 and 2643 K, respectively [8]. The high-temperature phases have an important role in structural and oxygen ion conduction, and they can be stabilized at room temperature through the substitution of Zr^4+^ with rare earth or alkali earth cations [9]. The solid solution is formed as di- or tri-valent metal oxides are substituted for Zr^4+^ in zirconia. At that time, the vacancies of O^2−^, which are necessary for the thermally activated conduction process, were formed to maintain electrical neutrality. The conductivity increases because of concentration of oxygen vacancies increases with increasing dopant content. However, after the highest value is reached, conductivity begins to decrease regularly as the dopant content increases [10].

Zirconia-based electrolytes are stabilized by doping zirconia with binary oxides such as CaO, MgO, Sc_2_O_3_, and Y_2_O_3_. Among them, yttria-stabilized zirconia (YSZ) has high ionic conductivity at high temperatures and is the most widely studied as a zirconia solid electrolyte [11,12]. It is well known that the conductivity of YSZ decreases during aging at high temperatures (approximately 1000 °C). Moghdam et al. [13] reported that the precipitation of the tetragonal phase in a cubic matrix decreased the conductivity of YSZ. Kondoh et al. [14] suggested the oxygen ion vacancies 
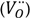
 around dopant cations (Y^3+^) are trapped in the dopant cations owing to Columbic forces, that is, short-range ordering. In addition, it has been reported that the grain boundary resistance increases due to glassy phase segregation and Y_2_Zr_2_O_7_ or Y_4_Zr_3_O_12_ phase precipitation by the formation of oxygen vacancy complexes [15,16]. YSZ is deteriorated by various factors, as described above, resulting in a decrease in conductivity. Many studies on the degradation mechanism have been conducted at temperatures below 1000 °C, whereas studies of the aging behavior at high temperatures above 1200 °C are required.

We studied heat stress-induced degradation behavior of 8 mol% Y_2_O_3_ stabilized zirconia (8YSZ) at 1300, 1400, and 1500 °C in terms of phase transformation, local atomic structure, and conductivity. XRD analysis is performed to confirm the phase transformation induced by aging, and the change in the local atomic structure is analyzed by the extended X-ray absorption fine structure (EXAFS) measurements. The changes in the electrical conductivity and activation energy with aging are investigated using impedance spectroscopy.

## 2. Experimental

8YSZ (TZ-8Y, Tosoh Co., Tokyo, Japan) was uniaxially pressed at 3 ton/m^2^ to produce 20 mm disk specimens. The green bodies were sintered at 1600 °C with a heating rate of 5 K/min and held at this temperature for 6 h. Sintered specimens were polished to a thickness of 1 mm and were heated in a tube furnace at 1300, 1400, and 1500 °C in atmospheric pressure in air for 100 h. The X-ray diffraction (XRD) patterns of the degraded specimens were collected at room temperature using a step scan procedure (2θ = 10–90°, step interval: 0.02°, Cu-Kα radiation, Rigaku Ultima-IV, Rigaku), and extended X-ray absorption fine structure (EXAFS) experiments were performed at the Zr K-edge using the EXAFS facility of the 10C wide-XAFS beamline in the Pohang Accelerator Laboratory (PLS-II, Pohang, Korea). The storage ring was operated at 3.0 GeV with an injection current of 350 mA. The sample surfaces were observed using field-emission scanning electron microscopy (FESEM, MIRA3, TESCAN). The grain size was determined from the SEM images based on ISO 13383—1 [17]. The grain size was defined as the diameter of the circumscribed circle from the image analysis. AC impedance measurements were performed with an Ivium-Stat (Ivium, Eindhoven, The Netherlands) instrument in the frequency range of 10^6^ Hz to 10^−2^ Hz using an excitation voltage of 10 mV, at an operating temperature of 1000 °C, in air.

## 3. Results and Discussion

SEM images of 8YSZ aged at 1300, 1400, and 1500 °C for 100 h (hereinafter termed 1300 °C 8YSZ, 1400 °C 8YSZ, and 1500 °C 8YSZ) are shown in Figure 1. The grain size increased from 10.53 μm to 25.81 μm by thermal aging, and the average grain sizes for 10 grains of the 8YSZ specimen at 1300, 1400, and 1500 °C were 20.83, 22.17, and 25.81 μm, respectively. This shows that the grain size gradually increased according to the thermal aging.

Figure 2 shows the XRD patterns of the aged 8YSZ. The (400) peak shifted to a higher angle for the 8YSZ specimens treated at 1300 °C and 1400 °C, and no new peak was observed. It was considered that lattice distortion due to the formation of a solid solution by the yttrium ions, which are larger than the zirconium ions, was relaxed by aging [18]. On the other hand, the peak shifted to a lower angle for the 1500 °C 8YSZ compared to 1300 °C and 1400 °C 8YSZ. It is related to the tetragonal peak observed at 1500 °C 8YSZ; the lower angle shift occurred with the formation of the tetragonal phase, which has a larger lattice [19].

The Zr K-edge Fourier transform analysis using EXAFS was performed to confirm the degradation behavior according to thermal aging (Figure 3); EXAFS signals were obtained in the range of 3 < K < 11.5 Å using a Hanning window. The first peak at 1.7 Å is attributed to Zr-O bonding, and the second peak at 3.31 Å appeared with the Zr-Cation (Zr, Y) [20,21]. The interatomic distance from the Zr ion to the first nearest neighbor was decreased in 1300, 1400 °C 8YSZ by degradation. This suggests that the lattice distortion caused by the substitution of Y^3+^ ions is larger than that of Zr^4+^ ions and is relaxed by aging, resulting in short-range ordering of the oxygen vacancies [22]. However, for 1500 °C 8YSZ, the interatomic distance to the first nearest neighbor of Zr increased with the formation of the tetragonal phase. As the heat stress increased, the distance to the second nearest neighbor was shortened, the intensity of the Zr-Cation peak decreased, and the intensity of the Zr-O peak increased. These results mean that the number of oxygen vacancies decreased due to the formation of a disordered phase by extraction of the cation dopant (Y^3+^). The increase in the intensity of the Zr-O peak for 1500 °C 8YSZ compared to that of 8YSZ treated at 1300 °C and 1400 °C corresponds to the previous XRD results [23].

Figure 4a shows the Nyquist plot from the impedance data for aged 8YSZ. The ohmic resistance (R*_ohm_*) appeared in the intercept on the real axis in the high-frequency region, and the electrode polarization resistance (R*_pol_*) was measured as the difference between the intercepts with the real axis at low and high frequencies [24]. R*_ohm_* and R*_pol_* increased as the heat stress increased, and the values of R*_pol_* were 6.82, 8.81, and 18.25 Ω cm^2^ for 8YSZ and the samples treated at 1300, 1400, and 1500 °C, respectively. The grain sizes increased with aging, but the area-specific resistance increased without decreasing. The increase in area-specific resistance by thermal degradation has a greater effect than the decrease in area-specific resistance due to the increase in grain size. Figure 4b shows the total ASR Arrhenius plots of 8YSZ before and after degradation. The activation energy of 1300 and 1400 °C 8YSZ was 0.86 and 0.87 eV, respectively, and the activation energy of 8YSZ at 1500 °C was significantly increased to 0.92 eV. It was considered that the short-range ordering hindered the diffusion of oxygen vacancies and increased the activation energy for 8YSZ treated at 1300 °C and 1400 °C, whereas for 1500 °C 8YSZ, the activation energy increased due to the formation of the tetragonal phase by Y^3+^ destabilization [25].

## 4. Conclusions

We investigated the degradation behavior of yttria-stabilized zirconia by thermal aging in terms of phase transformation, local atomic structure, and electrical conductivity. The average grain size of 8YSZ aged for 100h at 1300–1500 °C increased from 20.83 μm (1300 °C) to 25.81 μm (1500 °C). The cubic phase was predominant in all aged 8YSZ, and a higher angle shift of the (400) peak occurred with thermal aging at 1300 °C and 1400 °C 8YSZ, where no peak of the tetragonal phase was observed. For 1500 °C 8YSZ, the peak shifted to a lower angle due to the formation of a tetragonal phase with a large lattice. Zr K-edge EXAFS analysis of the Zr-O local atomic structure showed that the intensity of the Zr-O peak increased, and the intensity of the Zr-Cation peak decreased as the aging temperature increased. This implies that oxygen vacancies were reduced and Y^3+^ ions escaped from the lattice, resulting in 8YSZ destabilization. The interatomic distance from the Zr ion to the first nearest neighbor was decreased in 1300, 1400 °C 8YSZ, which is related to the short-range ordering of oxygen vacancies. However, for 1500 °C 8YSZ, the interatomic distance to the first nearest neighbor of Zr increased with the formation of the tetragonal phase. The area-specific resistance measured by impedance spectroscopy was increased as the heat stress increased. The activation energy of 1300 and 1400 °C 8YSZ was 0.86 and 0.87 eV, respectively, and the activation energy of 8YSZ at 1500 °C was significantly increased to 0.92 eV. The difference in the activation energy of the specimens according to the deterioration conditions is attributed to the decrease in oxygen vacancies and the increase in lattice distortion with the formation of the tetragonal phase.

## Figures and Tables

**Figure 1 materials-15-06947-f001:**
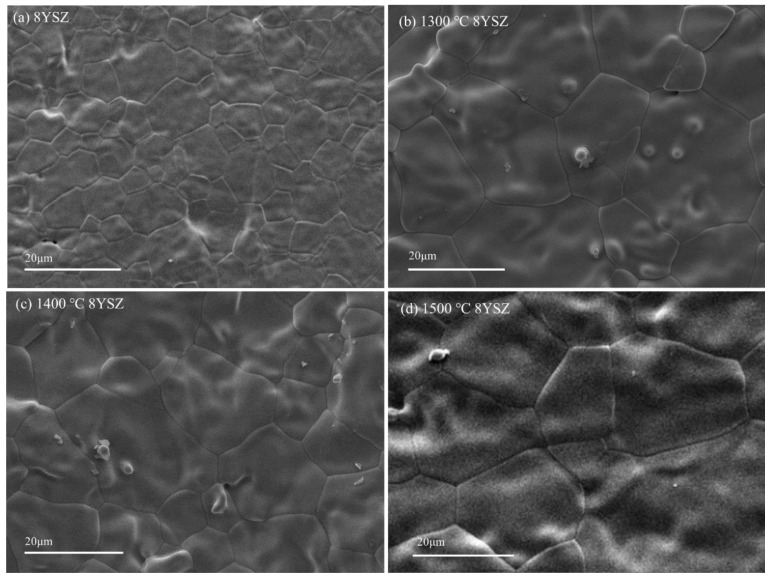
SEM images of specimens: (**a**) 8YSZ, (**b**) 1300 °C 8YSZ, (**c**) 1400 °C 8YSZ, (**d**) 1500 °C 8YSZ.

**Figure 2 materials-15-06947-f002:**
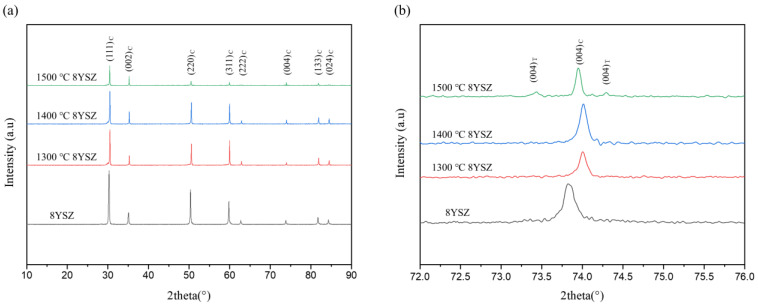
(**a**) XRD diffraction patterns and (**b**) (400) peaks of 8YSZ specimens after thermal aging for 100 h.

**Figure 3 materials-15-06947-f003:**
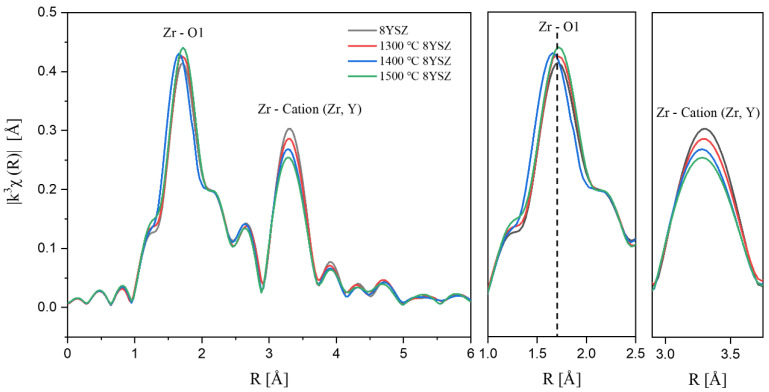
Zr K-edge Fourier-transform data for 8YSZ before and after thermal aging for 100 h.

**Figure 4 materials-15-06947-f004:**
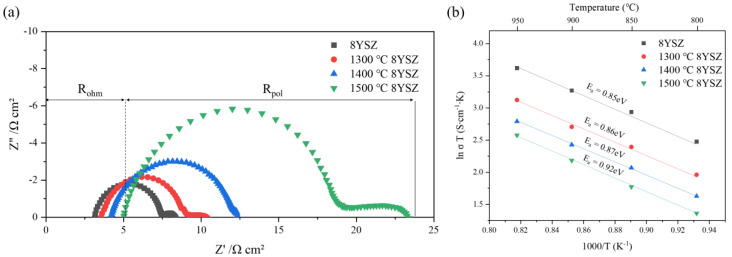
(**a**) Nyquist plot of 8YSZ before and after thermal aging; obtained in air at 1000 °C, (**b**) Arrhenius plots of the total ASR for 8YSZ before and after thermal aging.

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
