# Peer review of "Destabilization and Ion Conductivity of Yttria-Stabilized Zirconia for Solid Oxide Electrolyte by Thermal Aging"

_materials, 2022, doi:10.3390/ma15196947_

Round 1

Reviewer 1 Report

The authors investigate the thermal degradation of the the solid electrolyte yttria-stabilized zirconia by measuring XRD, fine structure, and conductivity. The conclusions are sound and have implications for use of the electrolyte at high temperatures.

I have the following suggestions:

In lines 60-62, the sintering and temperature stressing conditions should be more detailed. Was the oxide disk sintered and heated in air?

In Figure 2, the plot legend and labels are too small and should be made more readable.

In Figure 4, the plot legend and labels are too small and should be made more readable.

Author Response

Thank you for compliments and comments to the manuscript. Your sincere reviews have been very helpful to improve the quality of this paper.

Below is our reply to your comments.

[Reviewer’s comments]

Q 1) In lines 60-62, the sintering and temperature stressing conditions should be more detailed. Was the oxide disk sintered and heated in air?

You will see that a number of general and specific points are mentioned which necessitate extensive rewriting of the paper. You will see that a number of general and specific points are mentioned which necessitate extensive rewriting of the paper.

Answer) Thank you for the comments. We modified experimental to explain sintering and temperature stressing conditions in more detail.

- Before

“The green body was sintered at 1600 °C for 6 h and polished to a thickness of 1 mm. The polished specimens were heated in a tube furnace at 1300, 1400, and 1500 °C for 100 h.”

- After

“The green body were sintered at 1600 ℃ with heating rate of 5 K/min and held at this temperature for 6 h. Sintered specimens were polished to a thickness of 1 mm and were heated in a tube furnace at 1300, 1400, and 1500 ℃ in atmospheric pressure in air for 100 h.”

----------------------------------------------------------------------------------------------------------------

Q 2) In Figure 2, the plot legend and labels are too small and should be made more readable.

 In Figure 4, the plot legend and labels are too small and should be made more readable.

Answer) As the reviewer mentioned, we modified the plot legend and labels of Figure 2 and 4 to make readable.

Reviewer 2 Report

The work reports the destabilization and ion conductivity of yttria-stabilized zirconia for solid oxide electrolyte under thermal stress. The authors written the manuscript well and the results are interesting. However, the manuscript lacks a strong discussion of results.  Authors must address the following queries before accepting for publication in Materials. 

1. What is 8YSZ, authors should describe it at the first appearance.

2. The abstract needs to be rewrite in scientific manner.

3. The experimental sections should be clear, how the authors produced YSZ powder.

4. How the authors measured the average grain size from SEM analysis.

5. Authors reported that the (4 0 0) peak was shifted towards lower side for 1500 oC, but I did not found such shifting in Fig. 2(b).

6. Compare the XRD data with JCPDS data and indicate all the diffraction peaks with miller indices. 

7. The main concept of the study is the effect of thermal stress, unfortunately I did not found any discussion related to thermal stress in the manuscript.

8.  In EXAFS analysis, the authors reported that the interatomic distance to the first nearest neighbor is decrease for 1300, 1400 oC and increased for 1500 oC. I wondered that all the intensities at 1.7 and 3.31 Ao are increasing and decreasing order with respect to temperature respectively. How the authors confirmed it?

9. The impedance data shows the linear increase in both polarization resistance and activation energy. How to confirm the optimal temperature. Hence it is advised to do at higher temperature.

10. Conclusions need to be revised.

11. There are several grammatic errors, please verify the manuscript thoroughly. 

Author Response

Thank you for compliments and comments to the manuscript. Your sincere reviews have been very helpful to improve the quality of this paper.

Below is our reply to your comments.

[Reviewer’s comments]

Q 1) What is 8YSZ, authors should describe it at the first appearance.

You will see that a number of general and specific points are mentioned which necessitate extensive rewriting of the paper. You will see that a number of general and specific points are mentioned which necessitate extensive rewriting of the paper.

Answer) Thank you for your kind comment, we described 8YSZ at the first appearance.

--------------------------------------------------------------------------------

Q 2) The abstract needs to be rewrite in scientific manner.

Answer) According to the comment, we have faithfully revised the introduction in scientific manner.

--------------------------------------------------------------------------------

Q 3) The experimental sections should be clear, how the authors produced YSZ powder.

Answer) As the reviewer mentioned, we included more details of YSZ powder.

--------------------------------------------------------------------------------

Q 4) How the authors measured the average grain size from SEM analysis.

Answer) According to the comment, we explained how the average grain size were measured.

- Before

“The grain size was determined from the SEM images based on ISO 13383 – 1 [14].”

- After

“The grain size was determined from the SEM images based on ISO 13383 – 1 [14]. The grain size was defined as the diameter of the circumscribed circle form image analysis.”

--------------------------------------------------------------------------------

Q 5) Authors reported that the (4 0 0) peak was shifted towards lower side for 1500 oC, but I did not find such shifting in Fig. 2(b).

Answer) Thank you for the comments. We revised sentence for a more accurate explanation as follows.

- Before

“However, the peak shifted to lower angle for the 1500 °C 8YSZ.”
- After

“On the other hand, the peak shifted to lower angle for the 1500 ℃ 8YSZ compared to 1300 ℃ and 1400 ℃ 8YSZ.”

--------------------------------------------------------------------------------

Q 6) Compare the XRD data with JCPDS data and indicate all the diffraction peaks with miller indices.

Answer) According to the comment, we indicated all the diffraction peaks with miller indices.

--------------------------------------------------------------------------------

Q 7) The main concept of the study is the effect of thermal stress, unfortunately I did not find any discussion related to thermal stress in the manuscript.

Answer) Thank you for your kind comment. We found that the terminology can be confusing, so the word ‘thermal stress’ changed to ‘thermal aging’.

- Before

“Destabilization and ion conductivity of yttria-stabilized zirconia for solid oxide electrolyte under thermal stress”
- After

“Destabilization and ion conductivity of yttria-stabilized zirconia for solid oxide electrolyte under thermal aging”

--------------------------------------------------------------------------------

Q 8) In EXAFS analysis, the authors reported that the interatomic distance to the first nearest neighbor is decrease for 1300, 1400 oC and increased for 1500 oC. I wondered that all the intensities at 1.7 and 3.31 Å are increasing and decreasing order with respect to temperature respectively. How the authors confirmed it?

Answer) Thank you for your kind the comment. We modified Figure 4 to explain clearly.

The intensity of Zr-Cation peak was gradually decreased as the heat stress increased, which means extraction of the cation dopant (Y3+). The intensity of the Zr-O peak was gradually increased, which means that oxygen vacancies were reduced, and it is related to the extraction of the cation dopant (Y3+).

----------------------------------------------------------------------------------------------------------------

Q 9) The impedance data shows the linear increase in both polarization resistance and activation energy. How to confirm the optimal temperature. Hence it is advised to do at higher temperature.

Answer) Thank you for the kind comments. As the reviewer mentioned, it is reasonable to perform at higher temperature to confirm the optimal temperature. In general, the maximum temperature for measuring impedance is known to be under 1000 ℃, and the measuring equipment we have was also measured in this range because of measurement limitation. Therefore, if equipment capable of measuring at higher temperatures is established in the future, we plan to proceed as a further study.

In this study, we used AC impedance measurements to investigate the degradation behavior of yttria-stabilized zirconia by thermal aging. Since we were able to measure the degradation behavior of 8YSZ by measuring the activation energy and polarization resistance under the experimental conditions, it is considered that the measurement conditions for the impedance data are appropriate.

--------------------------------------------------------------------------------

Q 10) Conclusions need to be revised.

Answer) As the reviewer mentioned, we faithfully revised the conclusions.

--------------------------------------------------------------------------------

Q 11) There are several grammatic errors, please verify the manuscript thoroughly.

Answer) Thank you for your kind the comment. We tried to revise the manuscript of the grammatical errors and badly constructed sentences for improving a readability. The revised manuscript was received an Elsevier editing service.

Round 2

Reviewer 2 Report

Authors addressed all the queries to my concern. I recommend the manuscript to accept for publication.